# Ensemble inference of unobserved infections in networks using partial observations

**Renquan Zhang**[1], **Jilei Tai**[1], **Sen Pei**[2]*

**1** School of Mathematical Sciences, Dalian University of Technology, Dalian, China, **2** Department of Environmental Health Sciences, Mailman School of Public Health, Columbia University, New York, New York, United States of America

* sp3449@cumc.columbia.edu

**Data Availability Statement:** C++ codes for the ensemble inference algorithm and the modified DMP algorithm are available at the GitHub repository: https://github.com/SenPei-CU/EnsembleInference. Network data were

## Abstract

Undetected infections fuel the dissemination of many infectious agents. However, identification of unobserved infectious individuals remains challenging due to limited observations of infections and imperfect knowledge of key transmission parameters. Here, we use an ensemble Bayesian inference method to infer unobserved infections using partial observations. The ensemble inference method can represent uncertainty in model parameters and update model states using all ensemble members collectively. We perform extensive experiments in both model-generated and real-world networks in which individuals have differential but unknown transmission rates. The ensemble method outperforms several alternative approaches for a variety of network structures and observation rates, despite that the model is mis-specified. Additionally, the computational complexity of this algorithm scales almost linearly with the number of nodes in the network and the number of observations, respectively, exhibiting the potential to apply to large-scale networks. The inference method may support decision-making under uncertainty and be adapted for use for other dynamical models in networks.

## Author summary

Mathematical models in networks can be used to infer unobserved infections and support decision-making in outbreak control and management. However, a major challenge in the operational use of these models is model misspecification—whether models can reliably represent the real-world disease transmission process. As we can only observe part of the transmission process, mathematical models are inherently misspecified due to the uncertainty in the model structure and parameters. A reliable modeling method, therefore, should be robust to such uncertainty and imperfect knowledge of the transmission dynamics to support real-time decision-making. In this study, we used an ensemble inference method to infer unobserved infections using partial observations. We considered the condition where individuals have differential but unknown transmission rates as an example of model uncertainty mirroring the situation encountered in real-world settings. We found the ensemble inference can robustly infer unobserved infections using partial observations with uncertainty in model parameters, although the distribution of

downloaded from publicly available repositories. Links to these repositories are provided here: 1. Social contact network (http://www.sociopatterns.org/datasets/infectious-sociopatterns). 2. Hamsterster friendship network (http://konect.cc/networks/petster-hamster-household). 3. Scientific collaboration network (https://networks.skewed.de/net/new_zealand_collab). 4. US power grid (http://konect.cc/networks/opsahl-powergrid). 5. Road transportation network (http://konect.cc/networks/tntp-ChicagoRegional). 6. Slashdot friend network (https://www.aminer.cn/data-sna#Slashdot-small). 7. Digg reply network (http://konect.cc/networks/munmun_digg_reply). 8. Twitter following network (http://www.konect.cc/networks/ego-twitter). 9. Internet P2P network (http://www.konect.cc/networks/p2p-Gnutella31).

**Funding:** This work was supported by Liaoning Provincial Natural Science Foundation (Grant No.2022-MS-152 to RZ), Fundamental Research Funds for the Central Universities (DUT22LAB305 to RZ), Dalian High-Level Talent Innovation Program (Grant No.2020RQ061 to RZ), National Key Research and Development Program of China (2020YFA0713702 to RZ; 2021ZD0112400 to RZ), Centers for Disease Control and Prevention (U01CK000592 to SP; 75D30122C14289 to SP), National Science Foundation (DMS-2229605 to SP), and Council of State and Territorial Epidemiologists (NU38OT00297 to SP). The funders had no role in study design, data collection and analysis, decision to publish, or preparation of the manuscript.

**Competing interests:** The authors have declared that no competing interests exist.

parameters is mis-specified. The ensemble inference algorithm presents a general framework for inference of spreading dynamics in networks and may contribute to the toolbox supporting operational epidemic control in the face of model uncertainty and limited observation.

## Introduction

Emerging and recurrent infectious diseases are a perpetual threat to human society [1]. For most human-to-human transmitted diseases, effective control requires timely isolation of infectious individuals [2]. In reality, however, only partial infected individuals can be observed due to limited surveillance resources or asymptomatic and pre-symptomatic shedding. For instance, less than 10% of COVID-19 infections were reported in the US during the spring of 2020 [3–6]; in hospital settings, most patients carrying antimicrobial resistant bacteria cannot be identified unless they develop clinical symptoms [7]. Unobserved infections play an essential role for the sustain, growth and persistence of several infectious diseases. Evidence indicates that undocumented infections drove the early transmission of SARS-CoV-2 in China [8], and undetected carriers of methicillin-resistant Staphylococcus aureus, a prevalence drug-resistant pathogen, caused 75% of transmission events in two UK hospitals [9]. Modeling studies further demonstrate that intervention measures ignoring unobserved infections may not be able to contain outbreaks due to extensive stealth transmission [10–13]. As a result, identifying unobserved infections using sparsely observed data is critical for disease control.

Epidemic models in networks (or agent-based models) are frequently used to simulate infectious disease transmission [14]. Compared to compartmental models based on the assumption of homogeneous population mixing, agent-based models can capture the complex and heterogeneous person-to-person contact pattern. However, calibration of agent-based models to real-world data (i.e., inference of model parameters and states of individuals) is challenging [15]. Previous studies have addressed inference problems for epidemic models in networks. For example, given a snapshot of individual status at a time point, a centrality-based method was developed to infer the origin of an epidemic [16–18]. Later, more accurate algorithms based on message-passing and belief propagation were proposed to locate outbreak sources [19,20]. More recently, inference algorithms were coupled with agent-based models to assimilate multiple data streams to infer individual-level infection risk of COVID-19 [21,22] and antimicrobial resistant pathogens [23–25]. To address the issue of incomplete observations, data augmentation technique was used to augment missing data ensuring that the augmented data are consistent with observed data [26]. This technique has been coupled with Bayesian methods (e.g., Markov chain Monte Carlo) to reconstruct transmission chains of infectious diseases using epidemiologic, contact, and genomic data [27–30].

The problem of inferring missing infections has also been extensively studied in computer science and physics literature. An important topic is to learn the propagation networks of a spreading process using full or partial observations. Several studies developed inference methods based on likelihood maximization of the observed cascades depending on specific transmission models and model parameters [31–34]. Another line of research is to use Steiner tree sampling to reconstruct transmission networks under different observation models for node infection times [35–38]. This approach does not require assumptions on the transmission model and model parameters and can be used to estimate node infection probabilities. More generally, a dynamical message passing framework was developed to infer parameters of spreading models from partial observations [39]. Despite these efforts, inference of unobserved infections using sparse observations is still challenging in real-world settings.

Several challenges and practical considerations need to be addressed before putting inference systems into operational use. Successful inference using methods based on epidemic models requires a well-specified model that can reasonably represent the real-world disease transmission process. However, detailed information on the transmission process is typically not available in real-world applications. Models used in inference systems therefore do not necessarily capture the full complexity of realistic disease transmission dynamics. For instance, most previous studies used epidemic models with a constant transmission rate (i.e., the probability of infection for a contact between susceptible and infected individuals) to validate and compare inference algorithms. In contrast, real-world transmission rate can be highly heterogeneous due to a complex immunological landscape and other personal variations in susceptibility. For instance, vaccinated individuals are less likely to be infected following exposures to persons infected with influenza [40] and SARS-CoV-2 [41]. Ignoring this individual variation may lead to strong model misspecification and undermine inference accuracy. Further, sparse observations impose additional challenges for identifying unobserved infections. For operational use, a robust algorithm that can handle model misspecification and sparse observations is needed.

In this study, we use an ensemble Bayesian inference method to infer unobserved infections in networks using partial observations. Ensemble approach can handle uncertainty in model structure and was demonstrated to be able to support robust predictions in operational infectious disease forecasting (e.g., influenza and COVID-19) [42–44]. To make use of limited observations, we propagate information obtained from these observations to unobserved individuals in contact networks. We simulate epidemic outbreaks in networks using a modified susceptible-infected-recovered (SIR) model with heterogeneous transmission rates, for which the states of only part of individuals are observed. We evaluate the proposed inference algorithm on both synthetic and real-world networks and compare it with several alternative methods. Results indicate that the ensemble inference algorithm can better identify unobserved infections when the parameters of epidemic process are not perfectly known. This inference algorithm does not require specific forms of transmission dynamics and may be flexibly generalized to other epidemic models.

## Materials and methods

### Epidemic model

We used a modified SIR model in networks to simulate epidemic outbreaks. For an undirected network $G(V,E)$, where $V$ is the set of vertices and $E$ is the set of undirected edges, each node is in one of three states: susceptible (S), infected (I) and recovered (R). A susceptible individual $i$ can be infected by an infected neighbor with a transmission rate $\beta_i$ on each day, independent of the states of other neighbors. An infected individual recovers with a probability $1/D$, where $D$ is the average infectious period. Individual transmission rates were randomly drawn from a bimodal distribution, mimicking the situation where certain population are partially protected (see Fig A in S1 Text). Outbreaks were initiated by a set of randomly chosen seeds. In this study, we focus on the growth phase of epidemics, when timely inference can be used to inform interventions. Specifically, we selected configurations of initial seeds, transmission rate distribution, and simulation time so that the percentage of individuals ever infected is around 25–35% by the end of simulation (see Table A in S1 Text).

### Observation model

To model the real-world observation process, individuals were tested with a probability on each day depending on their states. We assume infected individuals are more likely to be tested than susceptible and recovered persons as infected people may have symptoms. In model

simulations, the percentage of total tested population ranged from 10% to 55%. Given the low daily testing probability and the relative short simulation period (Table A in S1 Text), most observed individuals were only tested once. These partial observations were used to support inference to locate unobserved infections. We also tested another observation model based on contact tracing. Contact tracing were widely used to find infections for epidemic control [45]. In this model, each node was tested with a daily testing probability, similar to the previous observation model. However, once an individual was tested positive on day $t$, all neighbors of infection in the contact network were tested on the next day $t+1$. As observing neighbors of infections would increase the number of observations, we adjusted the daily testing probability so that the percentage of tested individuals was between 10% to 50%.

## Problem statement

Define $P(X_i^t)$ as the probability of individual $i$ in the state $X \in (S,I,R)$ on day $t$. For a subset of nodes in the network, their states on certain days were observed. Define the observation data as $\mathcal{D} = \{(i_o, t_o, O)\}$, where $(i_o, t_o, O)$ represents that individual $i_o$ was observed in the state $O \in (S,I,R)$ on day $t_o$. The objective is to estimate $P(I_i^t)$ for each node $i$ given partial observations of individuals' states $\mathcal{D}$: $P(I_i^t|\mathcal{D})$ for $i = 1, \cdots, N$, where $N$ is the total number of individuals in the network. We are particularly interested in the estimates of $P(I_i^T)$ on the last day $T$, which can be used to inform prospective interventions to control outbreaks.

## Inference

We used an ensemble Bayesian inference method to infer unobserved infections using sparse observations. The evolution of $P(X_i^t)$ is governed by a set of master equations:

$$\frac{dP(S_i^t)}{dt} = -P(S_i^t)\left[1 - \Pi_{j \in \partial i}(1 - \beta_i P(I_j^t))\right], \tag{1}$$

$$\frac{dP(I_i^t)}{dt} = P(S_i^t)\left[1 - \Pi_{j \in \partial i}(1 - \beta_i P(I_j^t))\right] - \frac{P(I_i^t)}{D}, \tag{2}$$

$$\frac{dP(R_i^t)}{dt} = \frac{P(I_i^t)}{D}, \tag{3}$$

Here, $\beta_i$ is the transmission rate of node $i$, $\partial i$ is the set of neighbors of node $i$, and $D$ is the average infectious duration. Eqs (1–3) are accurate when the network has a tree structure. For real-world networks without too many short loops, previous studies found Eqs (1–3) provide a good approximation [46].

Depending on the initial conditions and model parameters, an infinite number of possible configurations of $P(X_i^t)$ exist. The proposed method maintains an ensemble of $P(X_i^t)$ configurations, each of which represents one possible realization of $P(X_i^t)$. We initiated $P(X_i^{t=0})$ for each ensemble member on the first day by randomly assigning a small infection probability to each node. Within each ensemble member, the individual transmission rates $\beta_i$ were drawn from a uniform distribution over a plausible range. We use this ensemble to represent the uncertainty in heterogeneous transmission rates given imperfect knowledge of individuals' transmission rate. Note that the uniform distribution of $\beta_i$ in the inference system does not match the actual bimodal distribution in outbreak simulations; nevertheless, as shown later, the inference results are robust to this mis-specified setting.

We sequentially updated probabilities $P(X_i^t)$ in each ensemble member using sparse observations. On each day $t$, three procedures were performed.

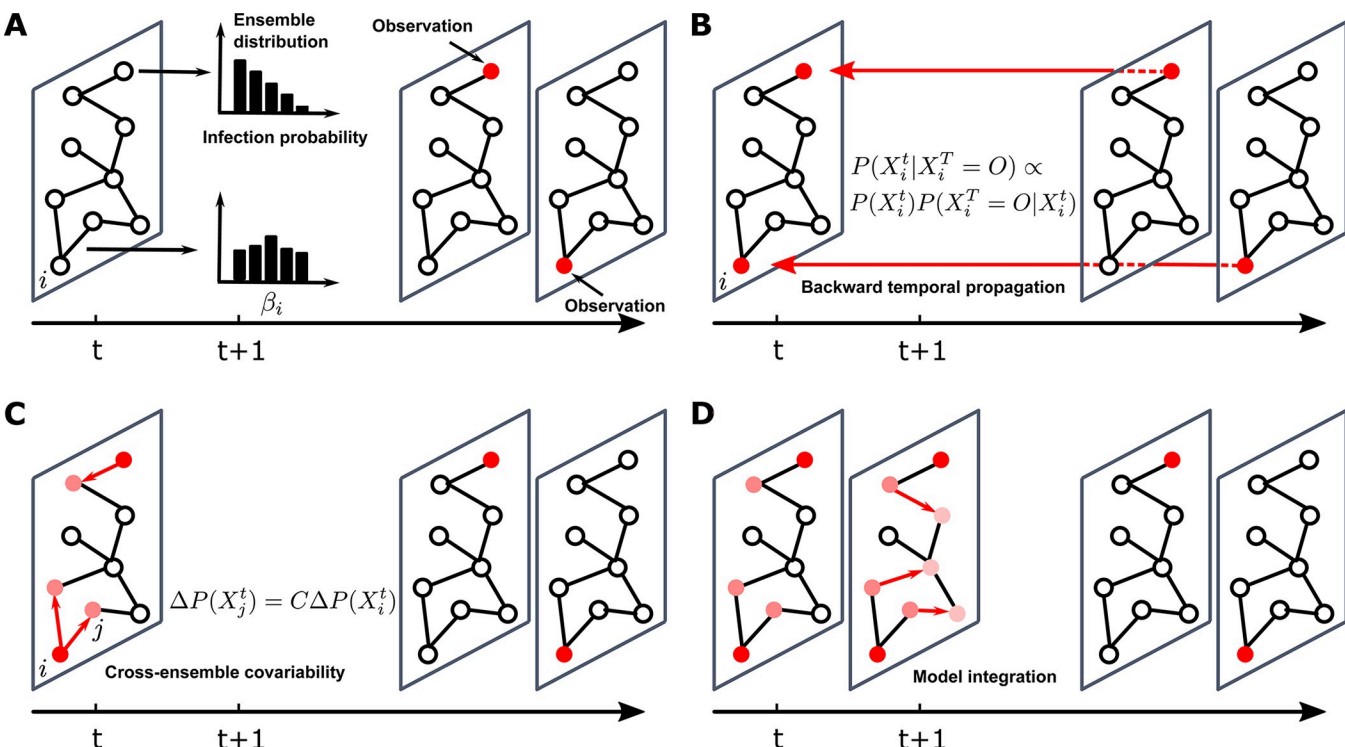

**Fig 1. An illustration of the ensemble inference algorithm. (A)**. Model state at time *t* and observations after time *t*. Two infected individuals are observed after time *t*. The red color represents the probability of infection. We use an ensemble of model states to represent the distribution of transmission rates and the distribution of infection probabilities. (**B**). We use Bayes' rule and master equations to propagate information backward in time and estimate the infection probability of observed individuals at time *t*: $P(X_i^t|X_i^T = O) \propto P(X_i^t)P(X_i^T = O|X_i^t)$, where node *i* is observed to be in state *O* at time *T*. (**C**). We use cross-ensemble covariability to adjust the infection probability of individuals connected to observed nodes. Denote the adjustment on observed node *i* as $\Delta P(X_i^t)$ for each ensemble member. The adjustement on node *j*, a neighbor of node *i*, is $\Delta P(X_j^t) = C\Delta P(X_i^t)$, where $C = cov(P(X_i^t), P(X_j^t))/var(P(X_i^t))$. Here $cov(P(X_i^t), P(X_j^t))$ is the covariance between $P(X_i^t)$ and $P(X_j^t)$ and $var(P(X_i^t))$ is the variance of $P(X_i^t)$, both computed using the ensemble members. Red arrows mean that the information on infection probability is propagated to the neighbors of observed nodes. (**D**). We integrate the master equations to time *t*+1. Red arrows show that information on infection probability is further propagated forward to neighbors. After these three procedures, information from the two observations reaches six other individuals in the network.

1. *Backward temporal propagation*: Assume individual $i_o$ is observed in state $O$ on day $t_{i_o} > t$ (Fig 1A). For each ensemble member, we estimate the probability of $i_o$ in each state $X \in \{S, I, R\}$ on day *t* given the future observation on day $t_{i_o}$, $P(X_{i_o}^t|X_{i_o}^{t_{i_o}} = O)$, using a Bayesian approach and the master equations (Fig 1B) (see details in S1 Text). We perform this procedure for all observations after day *t*. This procedure propagates information in future observations backward in time to the current day *t*.

2. *Cross-ensemble covariability adjustment*: We update the probabilities of $i_o$'s neighbors using cross-ensemble covariability between the probabilities for individual $i_o$ and neighbors (Fig 1C) (see S1 Text). The cross-ensemble covariability captures the dynamical coupling between $i_o$ and neighbors in the transmission dynamics and is directly computed using all ensemble members [36]. We perform this update for all observations after day *t*. This procedure propagates information from observed individuals to their neighbors on day *t*.

3. *Model integration*: We integrate the master equations using the updated probabilities from day *t* to day *t*+1, evolving $P(X_i^t)$ to $P(X_i^{t+1})$ for all individuals (Fig 1D). This procedure further propagates information in space (i.e., neighbors of nodes) and time.

We performed the above three procedures on each day sequentially until the most recent day with observations. We anticipate the information from partial observations can eventually reach the majority of individuals in the network. The mean estimate of $P(X_i^t)$ is the average value of all ensemble members. In this study, 100 ensemble members were used in the inference. Experiments with more ensemble members are reported in Results. The pseudo-code of the algorithm is provided in S1 Text. Implementation details of the ensemble inference algorithm are provided in S1 Text.

## Competing methods

A number of existing methods were used to rank the infection probability of individuals. (1). Degree. Nodes with more connections with other nodes are assumed to have a higher probability to be infected. (2). Contact. We rank nodes according to their number of connections to observed infections. Nodes with more exposure to observed infections are assumed to have a higher infection probability. (3). A modified dynamic message-passing (DMP) algorithm. The DMP algorithm uses a set of message-passing equations to quantify the evolution of probabilities of individuals' states. Initially used to identify the origin of an epidemic [19], the DMP algorithm was modified in this study to infer infection probability of each node. To represent the uncertainty in transmission rate, we implemented two versions of modified DMP algorithm. In the first, denoted as DMP1, transmission rates were set as a constant, the mean of a uniform distribution; in the second, denoted as DMP2, transmission rates for all individuals were randomly drawn from a uniform distribution, same as in the ensemble inference algorithm. We found the performance of these two approaches is similar. Details of the modified DMP algorithm are provided in S1 Text. We note that (1) Degree and (2) Contact have been used as baselines in many previous studies; however, as these strategies are widely used and are easy to implement in real-world settings, we decide to include them in experiments.

## Results

### Experiments in simulated networks

We first validated the ensemble inference algorithm using synthetic outbreaks on simulated networks. We compared the performance of the ensemble inference with several alternative methods to locate unobserved infections. These methods rank individuals' infection probability using the number of connections to other nodes (Degree), number of connections to observed infections (Contact), and a modified dynamic message-passing approach [19] (DMP1 and DMP2), respectively (see S1 Text). We applied each method to rank the infection probability of all nodes at the end of the simulation and used the receiver operating characteristic (ROC) curve to evaluate the performance [47]. Methods with a large area under the ROC curve (AUC) have a superior ability to infer unobserved infections.

We first validated the inference algorithm on trees, for which the master Eqs (1–3) are accurate. We ran an SIR model in a tree with 781 nodes and 780 edges. Each node has a maximum of 5 children and the maximum depth of the tree is 4. Transmission rates were drawn from a bimodal distribution (Fig A in S1 Text). The states of 18%, 36% and 55% individuals were observed at different time points. We used the partial observations to inform the ensemble inference algorithm to estimate the probability of each node to be infected on the last day of simulation. The ensemble inference algorithm performed better than other methods (Fig 2). In particular, the accuracy of top-ranking individuals was high, possibly due to the accurate master equations in tree structure. The DMP algorithm was proved to provide exact inference on trees to infer the origin of an epidemic [19]. For the inference task in this study, however,

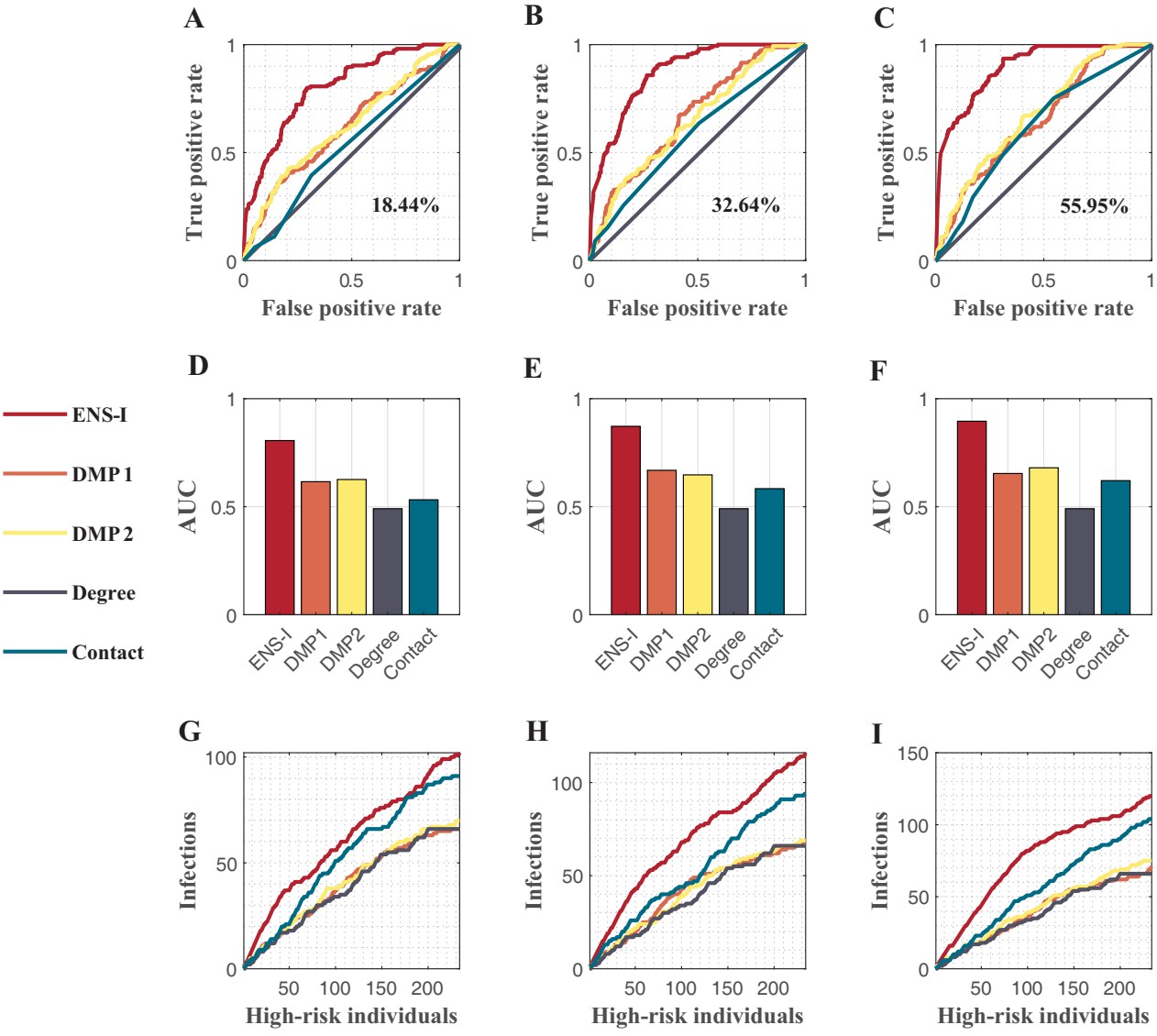

**Fig 2. Performance of different methods in a tree graph.** The tree has 781 nodes 780 edges. Five different methods, including the ensemble inference (ENS-I), modified dynamic message-passing with a fixed transmission rate (DMP1), modified dynamic message-passing with uniformly distributed transmission rates (DMP2), number of connections (Degree), and contact with observed infections (Contact), are compared. The ROC curves for various observation rates 18.44% (**A**), 32.64% (**B**), and 55.95% (**C**) are shown. The corresponding AUC values are compared in (**D**, **E**, **F**). The numbers of infections identified among high-risk individuals selected by different methods are shown in (**G**, **H**, **I**).

the performance of the modified DMP algorithm degraded likely caused by the incompatible observation model and the difficulty solving DMP equations using sparse observations (see a detailed discussion on the performance of the modified DMP algorithm in S1 Text).

We then validated the inference algorithm on homogeneous network. Homogeneous random networks were generated using the Erdős–Rényi (ER) model [48]. We generated an ER network with 1,000 nodes and an average degree of 2.6, ran an SIR model in the network for one week, and observed the states of around 15%, 30% and 50% of individuals at various time points. We estimate the infection probability of each node on the last day of simulation. Details on model configurations are provided in Table A in S1 Text.

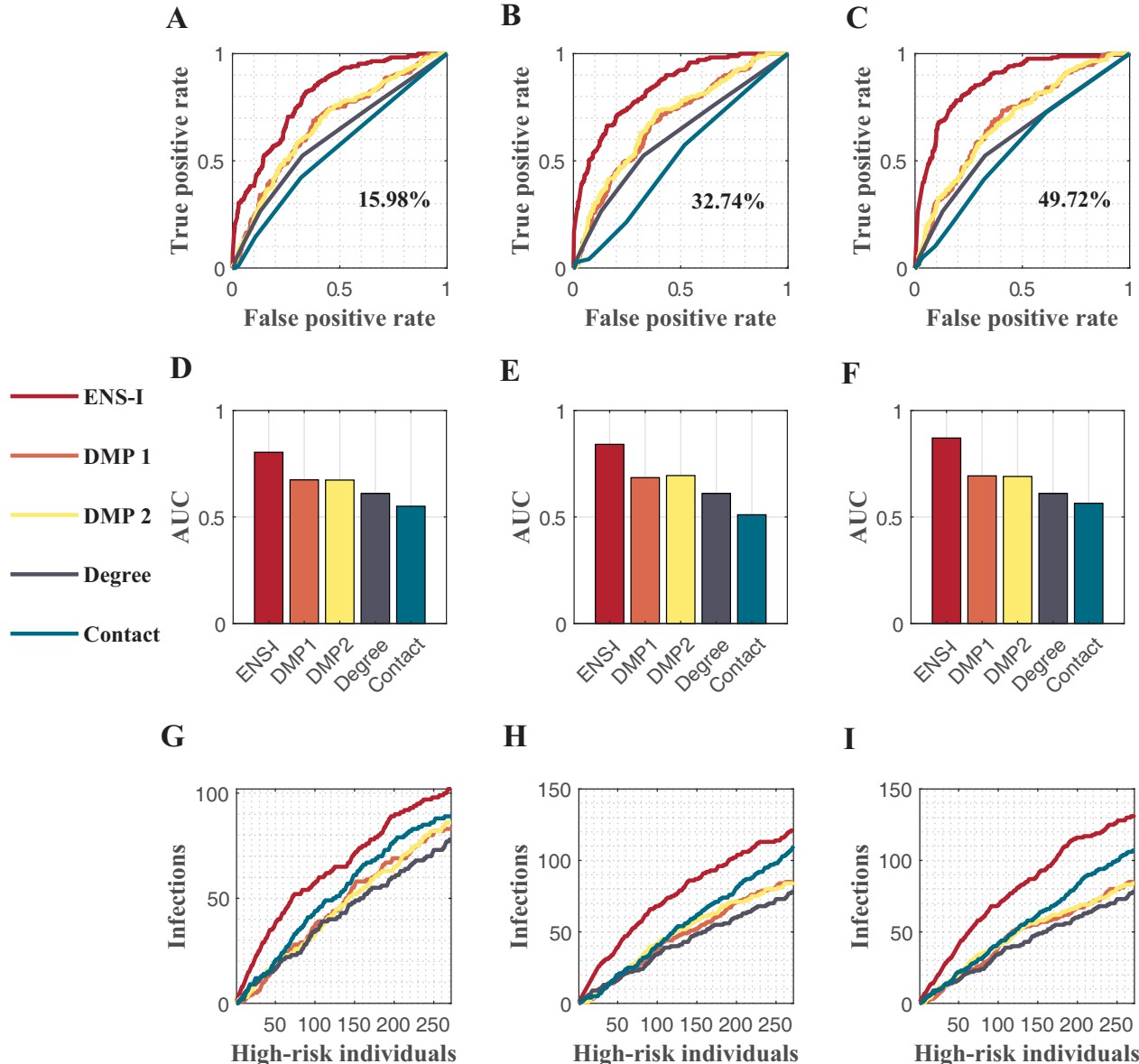

**Fig 3. Performance of different methods in homogeneous networks.** The ER network has 1,000 nodes with an average degree of 2.6. Five different methods, including the ensemble inference (ENS-I), modified dynamic message-passing with a fixed transmission rate (DMP1), modified dynamic message-passing with uniformly distributed transmission rates (DMP2), number of connections (Degree), and contact with observed infections (Contact), are compared. The ROC curves for various observation rates 15.98% (**A**), 32.74% (**B**), and 49.72% (**C**) are shown. The corresponding AUC values are compared in (**D**, **E**, **F**). The numbers of infections identified among high-risk individuals selected by different methods are shown in (**G**, **H**, **I**).

The ensemble inference algorithm outperformed other methods for the ER network with different observation rates (Fig 3A–3C). The modified DMP algorithm performed better than Degree and Contact, indicating that methods incorporated evolving dynamics are potentially more effective than metrics based purely on the network structure. The AUC for the ensemble inference was higher than other approaches (Fig 3D–3F). The ensemble inference was able to produce decent estimates of infection probabilities (AUC = 0.804) using observations from only 15% of the total population, even though the distribution of transmission rate was mis-specified. This experiment indicates that the ensemble inference algorithm is robust to the

uncertainty and variation in individuals' transmission rates. The accuracy of the ensemble inference algorithm increased with more observations; however, in general, the marginal benefit diminished with increasing observations (Fig 3A–3F). A possible reason for the diminishing marginal benefit may be that additional observations can contain overlapping information that already exists in current observations. For instance, observing a neighbor of an existing observation may not provide much additional information. This idea was shown in a study optimizing surveillance networks for respiratory diseases [49].

In real-world settings, typically, only a subset of individuals with higher infection probabilities are prioritized for testing given limited resources. As a result, the inference performance for the top-ranked individuals is more relevant for practical use. We compared the number of actual infections among the high-risk individuals selected by different approaches (i.e., how many infections can be found by screening the top-ranked individuals). The ensemble inference algorithm can identify more infections than other methods for a given number of estimated high-risk individuals (Fig 3G–3I).

Several additional experiments were performed to establish the effectiveness of the ensemble inference algorithm in homogeneous networks. The proposed algorithm outperformed competing methods in ER networks with a larger number of nodes and a different mean degree (Fig B in S1 Text), as well as when the transmission rate was drawn from a power-law distribution, representing the heterogenous capability to transmit disease (Fig C in S1 Text). The ensemble inference algorithm had a degraded performance in denser ER networks with a mean degree of 8 and 10 (Fig D in S1 Text), likely due to the lower accuracy of the master equations in dense networks. But the ensemble inference algorithm can still identify more infections in the top 10% higher-risk nodes, especially with more observations (Fig D in S1 Text). We ran the ensemble inference algorithm using 50, 100, 300, 500, and 1,000 ensemble members and found that the performance remained similar even if more ensemble members were used (Fig E in S1 Text, see a discussion in the Discussion section). We also ran experiments on a random regular graph in which all nodes have the same degree. The ensemble inference showed a better performance compared to others (Fig B in S1 Text).

A potential approach to further improve the performance of the modified DMP algorithm is to (1) estimate link-specific transmission rates using a dynamic message-passing algorithm [39] and (2) use estimated parameters along with the standard DMP algorithm to obtain estimates for the probability that a node is infected at a given time. However, inference of model parameters needs partial information on the activation times of nodes [39]. Observing activation times of nodes is difficult for the observation model used in this study and is in general challenging in real world due to the uncertain time period from infection to a positive test. To circumvent this difficulty, we assigned the actual transmission rates (or the actual transmission rates with a 10% random Gaussian noise) to the modified DMP algorithm. The performance of the modified DMP algorithm was still not satisfactory (Fig F in S1 Text), likely due to the observation model and issues solving the DMP equations using sparse observations.

We performed additional experiments in ER random networks using two other bimodal distributions for transmission rates, assuming different levels of protection in the population. In the first distribution, 20% individuals were randomly selected with lower transmission rates drawn from a Gaussian distribution $\mathcal{N}(0.18, 0.012)$ while other individuals had higher transmission rates drawn from $(0.22, 0.01)$. In the second distribution, 80% randomly chosen individuals had lower transmission rates ($\mathcal{N}(0.18, 0.012)$) and other individuals had higher transmission rates ($\mathcal{N}(0.22, 0.01)$). The ensemble inference algorithm outperformed other methods for both scenarios (Fig A in S1 Text).

We further tested another observation model based on contact tracing in ER random networks (see the Materials and Methods section). Contact tracing can discover infections among

the neighbors of infected individuals. Therefore, observed infections are not independent from each other. Nevertheless, the ensemble inference algorithm still achieved a satisfactory performance (Fig E in S1 Text).

Lastly, we performed experiments in scale-free (SF) networks. The SF network was generated using the configuration model [50], with a power-law degree distributions $P(k) \sim k^{-\gamma}$, where the power-law exponent $\gamma = 2.5$. The ensemble inference algorithm had a better performance than other methods (Fig 4).

## Experiments in real-world networks

Real-world networks have more complex structure than model-generated networks (e.g., community structure, clustering, assortative mixing, etc.). Such structural complexity can lead to

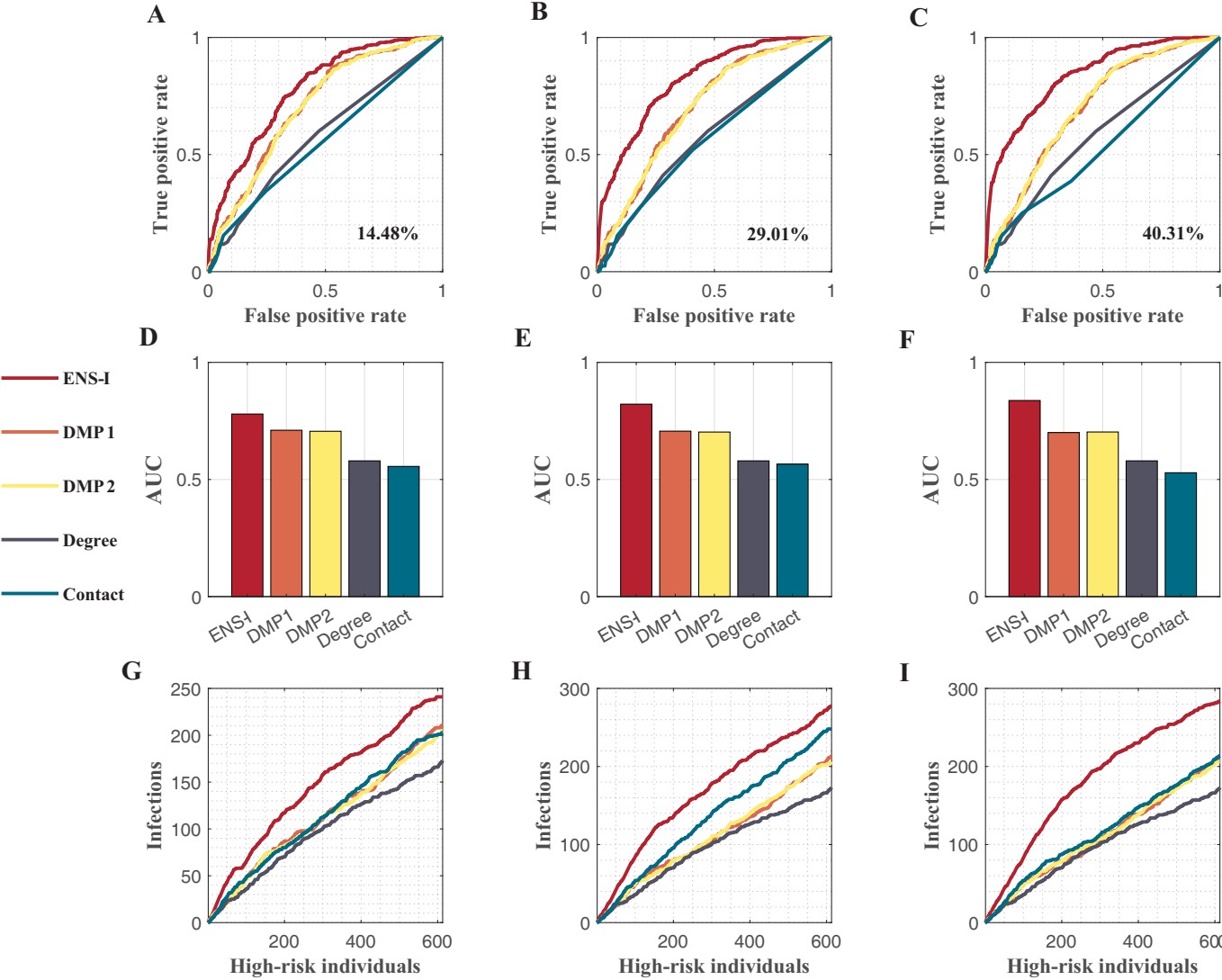

**Fig 4. Performance of different methods in scale-free networks.** The scale-free network has 2,044 nodes with a power-law exponent of 2.5. Five different methods, including the ensemble inference (ENS-I), modified dynamic message-passing with a fixed transmission rate (DMP1), modified dynamic message-passing with uniformly distributed transmission rates (DMP2), number of connections (Degree), and contact with observed infections (Contact), are compared. The ROC curves for various observation rates 14.48% (**A**), 29.01% (**B**), and 40.31% (**C**) are shown. The corresponding AUC values are compared in (**D**, **E**, **F**). The numbers of infections identified among high-risk individuals selected by different methods are shown in (**G**, **H**, **I**).

rich behaviors of transmission dynamics. To validate the performance of the ensemble inference, we ran outbreak simulations in nine real-world networks, downloaded from open-source repositories [51–53]. These networks span a wide range of systems including contact and social networks (social contact and collaborations), infrastructure (road and power grid), technological systems (internet peer-to-peer (P2P) network), and online social networks (Hamsterster, Twitter, Digg, and Slashdot). A detailed description of the networks and experiment settings is provided in Table B in S1 Text.

The ensemble inference algorithm exhibited a robust performance across different networks–it outperformed other methods in most networks (Figs 5 and 6). The DMP algorithm had comparable performance with the ensemble inference in several networks (Hamsterster, P2P, Slashdot, and Digg), but the overall performance of the ensemble inference was better among all considered networks. The Contact method did not perform well, possibly due to limited observation of infections. Experiments with varying observation rates yielded similar results (Figs G and H in S1 Text).

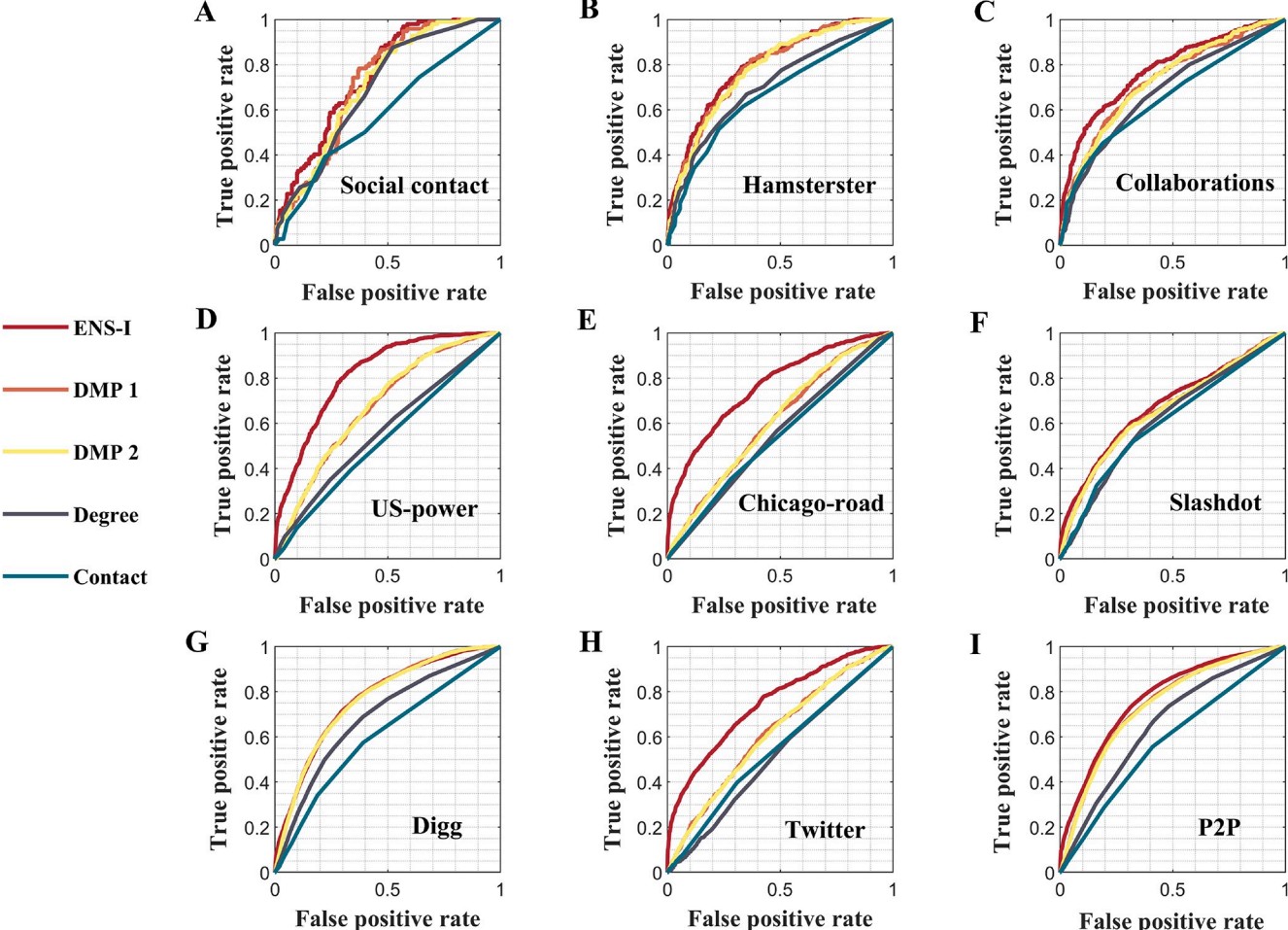

**Fig 5. Performance of different methods in real-world networks.** ROC curves for different methods (the ensemble inference (ENS-I), modified dynamic message-passing with a fixed transmission rate (DMP1), modified dynamic message-passing with uniformly distributed transmission rates (DMP2), number of connections (Degree), and contact with observed infections (Contact)) in nine real-world networks. The observation rates range from 12.64% to 17.61% in these networks. Details on the networks, model parameter setting, and observation rates are provided in S1 Text.

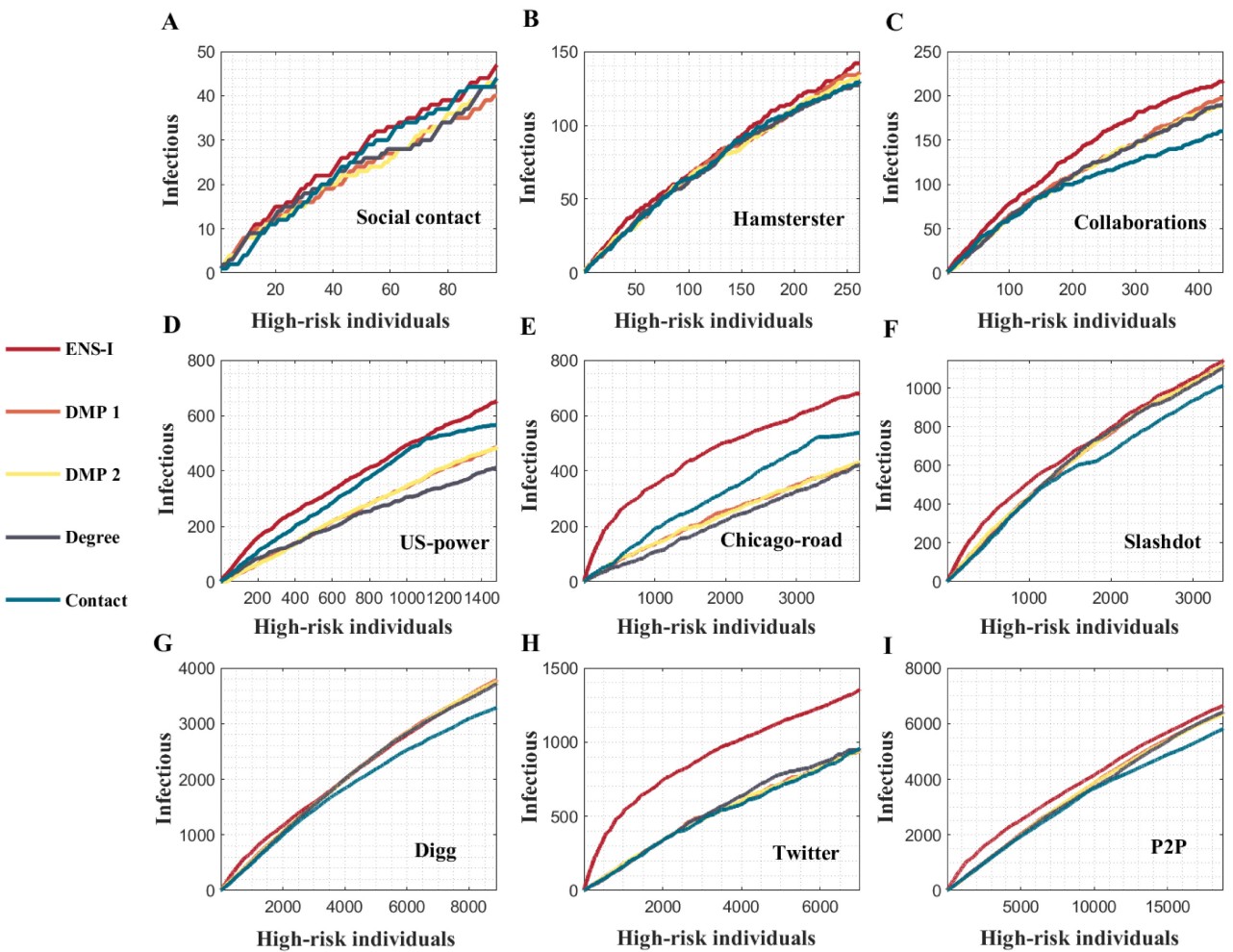

**Fig 6. Inference of high-risk individuals in real-world networks.** The numbers of infections identified among high-risk individuals selected by different methods are shown for nine real-world networks. Five different methods, including the ensemble inference (ENS-I), modified dynamic message-passing with a fixed transmission rate (DMP1), modified dynamic message-passing with uniformly distributed transmission rates (DMP2), number of connections (Degree), and contact with observed infections (Contact), are compared.

## Computational complexity

To examine the scalability of the ensemble inference, we ran the algorithm in ER networks with varying numbers of nodes, edges and observations. When the number of observations $N_o$ ($N_o = 200$) and the average degree $\langle k \rangle$ ($\langle k \rangle = 3.0$) were fixed, the running time $RT$ increased almost linearly with the number of nodes in the network $N$: $RT(N) \sim N^{1.144}$ (Fig 7A). For a fixed number of nodes ($N = 3,000$) and average degree ($\langle k \rangle = 3.0$), the computational time scaled linearly with the number of observations $N_o$: $RT(N_o) \sim N_o^{0.979}$ (Fig 7B). For a fixed number of nodes ($N = 3,000$) and observations ($N_o = 200$), the computational time increased sub-linearly with the average degree $\langle k \rangle$: $RT(\langle k \rangle) \sim \langle k \rangle^{0.172}$ (Fig 7C). If the percentage of observed nodes was fixed at 15% (i.e., $N_o = 0.15N$, $\langle k \rangle = 3.0$), the running time increased quadratically with the number of nodes: $RT(N) \sim N^{2.118}$ (Fig 7D). We also found that the computational time scaled almost linearly with the number of ensemble members $K$: $RT(K) \sim K^{0.973}$ when other settings remained the same ($N = 3,000$, $\langle k \rangle = 3.0$, $N_o = 200$) (Fig 7E). Given the polynomial computational complexity, it is possible to apply the ensemble inference algorithm to larger networks.

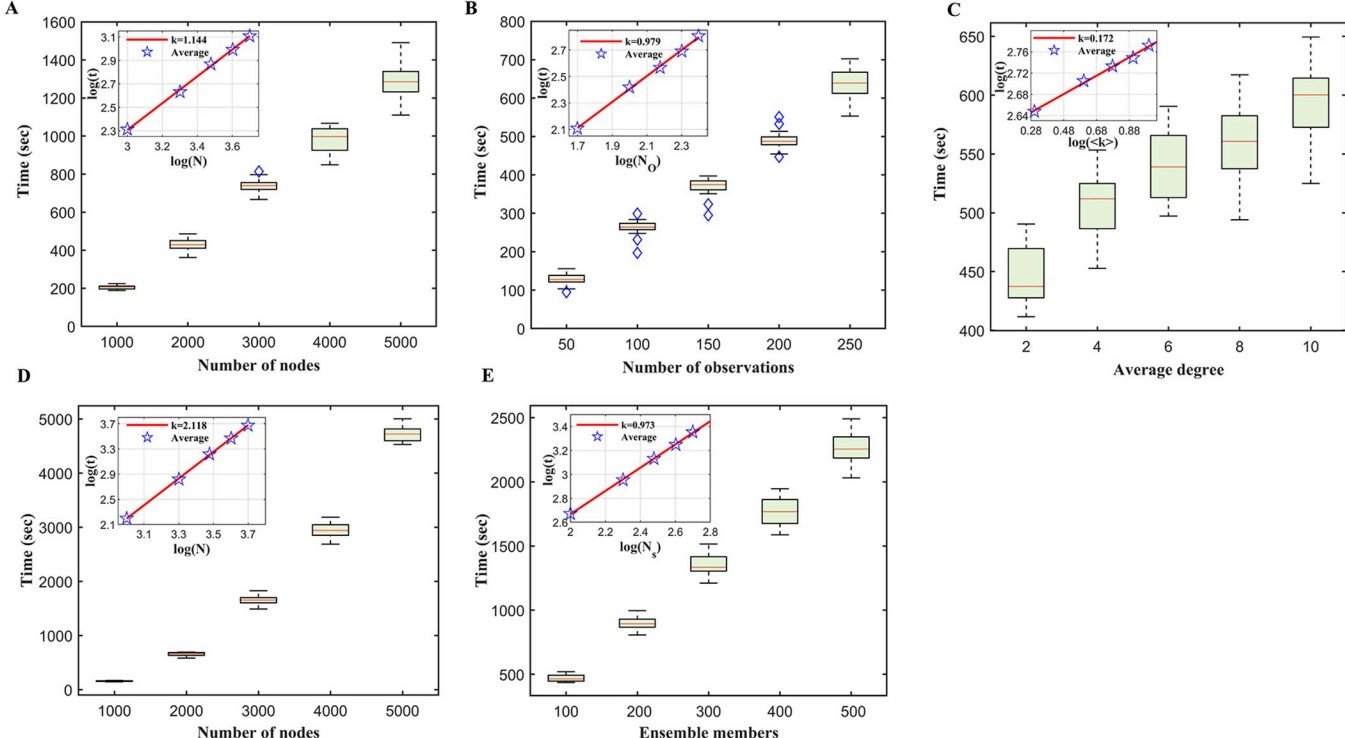

**Fig 7. Computational complexity of the ensemble inference algorithm.** (**A**). Computation time for increasing number of nodes with a fixed number of observations ($N_o$ = 200). Experiments were run on ER random networks with an average degree 3. Distributions of running time were obtained from 100 runs. Boxes show the median and interquartile. Whiskers show 1.5 times the interquartile range away from the bottom or top of the boxes. The inset shows the fitting of the computation time against the number of nodes (both log-transformed). (**B**). Computation time for a fix number of nodes 3,000 and a fix average degree 3 with increasing number of observations. (**C**). Computation time for a fix number of nodes 3,000 and a fix number of observations 200 with increasing average degrees. (**D**). Computation time for increasing numbers of nodes with a fixed percentage of observations (15%). (**E**). Computation time for increasing number of ensemble members in an ER random network with 3,000 nodes, an average degree of 3, and 200 observations.

In addition, the algorithm can be further expedited as the integration of master equations for different ensemble members can be run independently and in parallel.

## Discussion

The proposed inference method can be applied in several settings. For instance, during an outbreak of an emerging pathogen with a high rate of asymptomatic infection, partially observed infections coupled with detailed contact information can be used to support the inference of unobserved spreaders. In hospital settings, the inference method can be used to identify asymptomatic carriers of antimicrobial-resistant organisms based on sparse observations of clinical infections, limiting the nosocomial transmission of these pathogens.

The advantage of the ensemble inference under parameter uncertainty is possibly due to the way in which information is propagated across the network. We used cross-ensemble covariability to adjust the colonization probabilities of nodes connected to observed individuals. Such adjustment makes use of all ensemble members with various parameter settings [54–56] (as we need all ensemble members to compute the covariability), which prevents adjustment relying on only one parameter setting. In contrast, techniques such as message-passing propagate information using one set of pre-defined parameters [57,58]. If these parameters are close to the true values, message-passing-based methods can achieve very high accuracy as the evolution of infection probability is accurately described by the message-passing equations

[19,39,57–60]. However, in the case of unknown parameters, adjustment may lead to large errors if the pre-defined parameters are far from the truth. Moreover, such errors can be potentially further amplified by the nonlinear dynamics over successive adjustments across individuals in the network. The ensemble inference can avoid adjustments with extremely large errors by using an ensemble of model states with varying parameters. While the adjustment in ensemble inference may not be as accurate as in message-passing when the correct parameters are known, it does provide more robust inference given unknown parameters. In a recent study, ensemble inference methods were also found effective for models with noisy and expensive likelihoods [61].

Given the high dimensionality of the epidemic model, it is surprising that a few hundred ensemble members in the inference algorithm can yield a good performance. This could be due to several potential reasons. First, the purpose of using an ensemble of transmission rates for each edge is to avoid the large errors caused by using a single wrong parameter. As long as the ensemble can cover a reasonable range of edge-specific transmission rates, estimation errors will likely be limited and damped. In other words, the ensemble does not need to be a representative sample of all possible combinations of edge-specific transmission rates. Second, the ensemble of model states $P(X_i^t)$ for many individuals can be dynamically adjusted using covariablity with their neighbors. This "nudge" procedure can provide partial correction of model states so that the model can better fit observations. This phenomenon was also observed in the comparison of particle filter (or sequential Monte Carlo) [62] and ensemble Kalman filter [54,55]. In particle filter, the distribution of model states is represented by a weighted average of a large number of particles, each representing a potential combination of model parameters and variables. During model update, the weights of particles are adjusted but the model states remain the same (unless particles degenerate, i.e., dominated by a small number of particles, and needs to be re-sampled). In this case, it was found that the number of required particles grows exponentially with the number of parameters and variables [63], which limits its application to high-dimensional systems. In contrast, the ensemble Kalman filter, although suboptimal for nonlinear models, can be applied to high-dimensional systems such as weather models [54] and metapopulation epidemic models [64] using a relatively small number of ensemble members. The key reason is that ensemble members can be nudged according to observations, which creates more flexibility in model states. More studies are needed to explore the impact of ensemble size on inference performance.

Observation model is key to inference problems. The computational complexity and the nature of the inference problem change with the observation model. In fact, in many studies, observation model often determines the specific technical approach to solve problems. Several observation models were used in literature, including the infection times of all or partial nodes [19,31,35] and the states of all or partial nodes at a given snapshot [20,65,66]. In this study, we used a daily testing strategy to mimic tests among partial populations performed at various time points, which agrees with most real-world testing scenarios. We note that the inputs of the ensemble inference algorithm are the states of a fraction of population observed at different times (i.e., $\mathcal{D} = \{(i_o, t_o, O)\}$). As long as an observation model can provide such information, the ensemble inference algorithm is applicable. This offers the flexibility of the inference algorithm to work for different observation models.

There are a few limitations in this study. First, the master equations are accurate for tree-like networks without too many short loops. The accuracy of the inference dropped for dense networks with strong clustering. Further studied are needed to address this challenge. Second, the inference requires a plausible range of transmission rate, which may not be readily available during emerging outbreaks. The prior range of the transmission rate can be updated as

more epidemiological evidence is accumulated. Third, several practical obstacles need to be addressed before applying the inference in real world. For instance, real-world contact network data are difficult to collect except in certain settings (e.g., healthcare system where patients' movement is documented in electronic health records). A better representation of the contact network may improve the performance of the inference. Testing sensitivity and specificity were not considered in the experiments (we assumed a perfect testing) but should be incorporated into the inference system.

## Supporting information

**S1 Text. Supplementary materials.** Extended methods and analyses.
(PDF)

## Author Contributions

**Conceptualization:** Renquan Zhang, Sen Pei.

**Data curation:** Jilei Tai.

**Formal analysis:** Renquan Zhang, Jilei Tai, Sen Pei.

**Funding acquisition:** Renquan Zhang, Sen Pei.

**Investigation:** Renquan Zhang, Jilei Tai, Sen Pei.

**Methodology:** Sen Pei.

**Project administration:** Renquan Zhang, Sen Pei.

**Resources:** Renquan Zhang, Sen Pei.

**Software:** Jilei Tai.

**Supervision:** Renquan Zhang, Sen Pei.

**Validation:** Renquan Zhang, Jilei Tai, Sen Pei.

**Visualization:** Jilei Tai.

**Writing – original draft:** Sen Pei.

**Writing – review & editing:** Renquan Zhang, Jilei Tai.

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
