## [Decision Letter · Decision Letter 0]

17 May 2023

Dear Dr. Pei,

Thank you very much for submitting your manuscript "Ensemble inference of unobserved infections in networks using partial observations" for consideration at PLOS Computational Biology.

As with all papers reviewed by the journal, your manuscript was reviewed by members of the editorial board and by several independent reviewers. In light of the reviews (below this email), we would like to invite the resubmission of a significantly-revised version that takes into account the reviewers' comments.

We encourage you to construcitvely address referees' comments, in particular those from R#2. We look forward to receiving your revised manuscript.

We cannot make any decision about publication until we have seen the revised manuscript and your response to the reviewers' comments. Your revised manuscript is also likely to be sent to reviewers for further evaluation.

Sincerely,

Feng Fu

Academic Editor

PLOS Computational Biology

Virginia Pitzer

Section Editor

PLOS Computational Biology

We encourage you to construcitvely address referees' comments, in particular those from R#2. We look forward to receiving your revised manuscript.

Reviewer's Responses to Questions

**Comments to the Authors:**

Reviewer #1: In this study, the authors proposed an ensemble inference method to infer unobserved infections under the condition of unknown model parameters. This method was applied to an SIR model in networks and was validated in both synthetic and real-world networks. The performance of the inference method was shown to be better than several competing methods. The authors also examined the computational complexity of the algorithm, showing its potential for applications in large-scale networks. This study addressed an important challenge in epidemic inference in real-world settings – uncertainty in model parameters. This work makes a meaningful contribution to the literature on inference problems of epidemic processes in networks, with potential implications in real-world applications. However, I have several suggestions before accepting the manuscript.

1. In “Experiments on simulated networks”, numerical simulations were done only in ER and regular networks. Similar experiments should be performed in model-generated scale-free networks.

2. To better understand the algorithm, I suggest to move the “Pseudo-code and implementation” section in the SI to the section “Materials and methods” in the main text. Readers need more details on the algorithm implementation.

3. In the discussion section, it would be helpful to list a few application settings where the proposed algorithm can be used.

4. The format of references should be checked carefully, such as [1], [36]

Above all, it is a novel paper. I would suggest acceptance with a minor revision.

Reviewer #2: Summary:

The paper presents an ensemble Bayesian inference method to infer missing or unreported infections in networks using partial observations. The objective is to estimate the probability of unobserved infections and inform prospective interventions to control outbreaks. The proposed method is able to handle uncertainty in model structure and demonstrates robust predictions in infectious disease forecasting. The algorithm is evaluated on both synthetic and real-world networks and outperforms considered baselines.

The paper uses a network-based SIR model with individual transmission rates drawn from a bimodal distribution and a common recovery rate for infected individuals. To model the real-world observational process, individuals are tested with a probability on each day depending on their states. The ensemble Bayesian inference algorithm uses these partial observations to estimate the probability to estimate the probability of individuals being in a particular state (S, I, R) on each day of an outbreak, given partial observations. For each (node, state, time) tuple, the probability of the node being in that particular state at that time is the probability of interest. It is computed using master equations that account for network structure, time dependencies, and transmission & recovery rates.

The key novel contribution here is that the authors account for uncertainty in transmissibility rates using an ensemble of possible realizations of these probabilities and update them sequentially using the sparse observations, backward temporal propagation, cross-ensemble covariability adjustment, and model integration. Each member of the ensemble differs in the network edge probabilities, which are chosen randomly from a bimodal distribution. Throughout, 100 ensemble members were used in the inference.

The performance of the ensemble inference algorithm is compared to several baselines, including degree, contact, and modified dynamic message-passing approach (DMP1 and DMP2), which ranked individuals' infection probability. For the experiments, both synthetic and publicly available real-world networks have been used. Networks have been generated using the Erdős–Rényi (ER) model for the synthetic network experiments. The ensemble inference algorithm outperforms other methods, even when the distribution of transmission rate has been mis-specified. The accuracy of the ensemble inference algorithm increases with more observations, but the marginal benefit diminishes with increasing observations. For the inference on real-world networks, performance for the top-ranked (high-risk) individuals has been considered. The ensemble inference algorithm identifies more infections than other methods for a given number of estimated high-risk individuals. Additional experiments have been conducted to establish the effectiveness of the ensemble inference algorithm in homogeneous networks, including ER networks with a larger number of nodes, networks with transmission rate drawn from a power-law distribution, and random regular graphs.

Major comments:

Problem is not formally stated: The basic objective is to infer unobserved infections. The setup is provided in the "Inference" subsection in "Material and Methods". However, there is no formal problem statement where the set of observed infections is specified. It has to be understood from the section on epidemic model. Also, the observation model should be separated from the epidemic model as the method used for observation clearly can be applied to multiple epidemic models.

Number of configurations in the ensemble: The main novelty of this paper is

in the use of ensembles of configurations. Throughout, 100 members are used in the ensemble. Even if we pick from a small finite set of edge probabilities, for a moderate-sized network, the number of possible combinations of transmission probabilities is very large. The authors do not provide an explanation for why only 100 members are considered. The reviewer is of the opinion that 100 configurations might not be representative. Considering that this is a key idea in the paper, more experimental evaluation needs to done.

No theoretical guarantees: Theoretical treatment could have greatly strengthened this paper. Some results, even under the assumption of some restricted family of graphs, inference model, and model parameters could throw light on the ability and limitations of the proposed method. For example, in line with Lokhov et al. (2014), the authors could have attempted some result for trees: How does the ensemble-based algorithm perform on trees for which the master equations are accurate, and for which the DMP algorithm provides exact inference.

Literature survey: The authors have missed several publications on missing infections. This is an important topic that has been very well studied in epidemiology, computer science, and physics. Here are some examples.

Rozenshtein, P.; Gionis, A.; Prakash, B. A.; and Vreeken, J. 2016. Reconstructing an Epidemic Over Time. In Proceedings of the 22nd ACM SIGKDD International Conference on Knowledge Discovery and Data Mining, KDD ’16, 1835–1844. New York, NY, USA: Association for Computing Machinery. ISBN 9781450342322.

Xiao, H.; Aslay, C.; and Gionis, A. 2018. Robust Cascade Reconstruction by Steiner Tree Sampling. In 2018 IEEE International Conference on Data Mining (ICDM), 637–646.

Jang, H.; Pai, S.; Adhikari, B.; and Pemmaraju, S. V. 2021. Risk-aware Temporal Cascade Reconstruction to Detect Asymptomatic Cases : For the CDC MInD Healthcare Network. In 2021 IEEE International Conference on Data Mining (ICDM), 240–249.

Ritwick Mishra, Jack Heavey, Gursharn Kaur, Abhijin Adiga, Anil Vullikanti, Reconstructing an Epidemic Outbreak Using Steiner Connectivity, AAAI'23.

The following publication is also quite relevant.

Lokhov, Andrey. "Reconstructing parameters of spreading models from partial observations." Advances in Neural Information Processing Systems 29 (2016).

It uses a DMP-based inference algorithm to estimate model parameters from partial observations. The reviewer suggests that the authors use this work in the following manner to obtain a baseline: (1) Run this algorithm to obtain estimates of the model parameters and then (2) use these model parameters along with the standard DMP algorithm to obtain estimates for probability that a node is infected. Xiao et al. (2018) consider robust cascade reconstruction and have a similar framework as this paper.

Secondly, the baselines degree and contact with observed infections have used as baselines in many of the above-mentioned works. It is not surprising that the proposed algorithm does better than them.

A very specific observation model: Many inference models have been considered in the literature. The authors consider a random testing based inference model and their entire study is based on this model. This is important as the complexity and the nature of the problem changes with the observation model.

Discussion section could be strengthened: The first two paragraphs are merely summary that has already been covered in other parts of the paper. The third paragraph could benefit from references. The fourth paragraph needs more careful analysis. Since this is an experimental work, unless similar analysis is made for models such as SIS or SI, the authors cannot claim generality. If the authors had proven some rigorous theoretical guarantees, then may be they could have been extended to other models. It can be stated as future work though.

Networks: Some of the real-world networks are not representative in the context of epidemiology. Epidemiological applications include spread of diseases, rumor spread, or spread of malware. The authors evaluate their algorithms on transport and protein networks. Why were these networks chosen? No networks correspond to infectious disease epidemiology. There are networks on human interactions that are publicly available and have been applied for disease spread studies. See for example:

SocioPatterns: Cattuto, Ciro, et al. "Dynamics of person-to-person

interactions from distributed RFID sensor networks." PloS one 5.7 (2010):

e11596.

Also, the average degree of ER graphs is quite low. More experiments with denser graphs could strengthen the paper.

Computational complexity: An analysis of the computational complexity is presented with varying network size and number of observations. The analysis was conducted on ER graphs and is quite detailed. But it is not clear what the size of network means. A network's size depends not only on the number of nodes but also edges. This information is not provided in Figure 5. The complexity will also depend on the number of ensembles. This is not addressed.

Bimodal distribution: The authors state that the reason for choosing bimodal distribution is to mimic the situation where a subpopulation is partially protected. But they seem to have considered the specific case of 50% of the population (on an average) being protected. It would be interesting to know how the performance changes by varying the parameters of the bimodal distribution and how probabilities are assigned.

Expedited integration of master equations: What is the difference between ENS-I O and ENS-I? In the expedited ensemble inference (ENS-I), the ensembles are being processed independently. Does this have something to do with cross-ensemble covariance?

Speed-up: In the supplement, the following statement is made: For randomly observed sparse observations, we can assume the interdependency of their states is negligible and estimate the posterior of i_o ... While this speeds up computation of the likelihoods, it is not clear if this independence can be assumed for all scenarios. Firstly, synthetic graphs (on which experiments were conducted) like ER-graphs have low clustering coefficient and other "high-girth" properties. Combined with the random observations model used in this paper, it is possible that there is some minimum distance between observed nodes. This could be the reason for the good performance. More analysis in this regard will help as this would help in scaling up to large graphs. Secondly, it is not clear if this works with other more realistic observation models like contact-tracing, where observed nodes are within a constant number of hops from each other.

Minor:

Fig. 1c: It does not contain enough information about the concepts that the authors are trying to illustrate. It can be greatly improved. What are the blue and pink colors?

Why does marginal benefit diminish with observations? Some explanation could have helped.

**Have the authors made all data and (if applicable) computational code underlying the findings in their manuscript fully available?**

Reviewer #1: **No: **The data are available but I do not find the computational code

Reviewer #2: **No: **Code is not available. Data is publicly available.

PLOS authors have the option to publish the peer review history of their article (what does this mean?). If published, this will include your full peer review and any attached files.

Reviewer #1: No

Reviewer #2: No
---

## [Decision Letter · Decision Letter 1]

12 Jul 2023

Dear Dr. Pei,

We are pleased to inform you that your manuscript 'Ensemble inference of unobserved infections in networks using partial observations' has been provisionally accepted for publication in PLOS Computational Biology.

Best regards,

Feng Fu

Academic Editor

PLOS Computational Biology

Virginia Pitzer

Section Editor

PLOS Computational Biology

Reviewer's Responses to Questions

**Comments to the Authors:**

Reviewer #1: Authors have addressed all my concerns.

**Have the authors made all data and (if applicable) computational code underlying the findings in their manuscript fully available?**

Reviewer #1: Yes

PLOS authors have the option to publish the peer review history of their article (what does this mean?). If published, this will include your full peer review and any attached files.

Reviewer #1: No

---

## [Editor Report · Acceptance letter]

1 Aug 2023

PCOMPBIOL-D-23-00345R1 

Ensemble inference of unobserved infections in networks using partial observations

Dear Dr Pei,

I am pleased to inform you that your manuscript has been formally accepted for publication in PLOS Computational Biology. Your manuscript is now with our production department and you will be notified of the publication date in due course.

With kind regards,

Zsofi Zombor
